# Predictors of Recurrent High Emergency Department Use among Patients with Mental Disorders

**DOI:** 10.3390/ijerph18094559

**Published:** 2021-04-25

**Authors:** Lia Gentil, Guy Grenier, Helen-Maria Vasiliadis, Christophe Huỳnh, Marie-Josée Fleury

**Affiliations:** 1Department of Psychiatry, McGill University, 1033, Pine Avenue West, Montreal, QC H3A 1A1, Canada; lia.gentil@douglas.mcgill.ca; 2Douglas Hospital Research Centre, Douglas Mental Health University Institute, 6875 LaSalle Blvd, Montreal, QC H4H 1R3, Canada; guy.grenier@douglas.mcgill.ca; 3Centre Intégré Universitaire de Santé et des Services Sociaux du Centre-Sud-de-l’Île-de-Montréal, Institut Universitaire sur les Dépendances, 950 Louvain Est, Montréal, QC H2M 2E8, Canada; christophe.huynh.ccsmtl@ssss.gouv.qc.ca; 4Département Des Sciences de la Santé Communautaire, Université de Sherbrooke, Longueuil, QC J4K 0A8, Canada; helen-maria.vasiliadis@usherbrooke.ca; 5Centre de Recherche Charles-Le Moyne-Saguenay–Lac-Saint-Jean sur les Innovations en Santé (CR-CSIS), Campus de Longueuil-Université de Sherbrooke, 150 Place Charles-Lemoyne, Longueuil, QC J4K 0A8, Canada

**Keywords:** emergency department, recurrent high users, high users, mental disorders, predictors

## Abstract

Few studies have examined predictors of recurrent high ED use. This study assessed predictors of recurrent high ED use over two and three consecutive years, compared with high one-year ED use. This five-year longitudinal study is based on a cohort of 3121 patients who visited one of six Quebec (Canada) ED at least three times in 2014–2015. Multinomial logistic regression was performed. Clinical, sociodemographic and service use variables were identified based on data extracted from health administrative databases for 2012–2013 to 2014–2015. Of the 3121 high ED users, 15% (*n* = 468) were recurrent high ED users for a two-year period and 12% (*n* = 364) over three years. Patients with three consecutive years of high ED use had more personality disorders, anxiety disorders, alcohol or drug related disorders, chronic physical illnesses, suicidal behaviors and violence or social issues. More resided in areas with high social deprivation, consulted frequently with psychiatrists, had more interventions in local community health service centers, more prior hospitalizations and lower continuity of medical care. Three consecutive years of high ED use may be a benchmark for identifying high users needing better ambulatory care. As most have multiple and complex health problems, higher continuity and adequacy of medical care should be prioritized.

## 1. Introduction

A small proportion of patients make disproportionate use of emergency departments (ED) [1,2], contributing to overcrowding in ED [3,4], longer length of stay and longer wait time for treatment [5], as well as increased patient dissatisfaction [6] and higher health care costs [4,7]. According to a Canadian scoping review [8] that assessed 20 studies of frequent ED use (60% from the U.S.), between 4% (from a telephone survey of 800 ED users in Taiwan [9]) and 29% (from a New York City study of 205,139 patients using Medicaid data [10]) of ED users are high users [10], generally defined as those who make 3–4 ED visits yearly [11]. This group accounted for 12% (from a U.S. study using general hospital data of 47,349 ED users [12]) to 67% of total ED visits (from a U.S. study assessing 3141 patients with acute asthma based on data from 83 U.S. ED [13]). A minority of these high one-year ED users are “recurrent” high ED users, meaning that they make frequent ED visits over several consecutive years [8]. In the U.S., 4% of ED users in New York City receiving Medicaid were found to visits ED three or more times per year over a three-year period, while 2% visited ED at this rate over five years [10]. Another U.S. study using San Francisco hospital data reported that 38% of high ED users remained high users for two consecutive years, while 56% for a third year [12]. Recurrent high ED use is often associated with having mental disorders (MD), substance-related disorders (SRD) and physical health conditions [8]. One New York City study identified 72% of recurrent ED users with MD, 59% SRD, 49% co-occurring MD/SRD and 78% with severe chronic physical illnesses [10], suggesting a tendency for ED to substitute for continuous ambulatory care particularly in vulnerable populations. The influx of recurring high users in ED may also negatively impact on ED clinicians, casting doubt on their ability to treat patients [14] and further underlining the importance of reducing or preventing high and recurrent high ED use. 

Few studies have assessed predictors of recurrent high ED use in the general population [10,15,16,17,18] and among patients with MD more particularly [2,14,19]. Aside from high rates of MD [10,17], SRD [10,17,18], chronic physical illnesses [10,15,16,17] and co-occurring illnesses [10,18] among recurrent high ED users, studies have identified more women [17], individuals living in areas with high social deprivation [17], those in more frequent contact with psychiatric [15] or other ambulatory care services [10,16] as well as evidence that having a general practitioner (GP) may prevent recurrent high ED use [17]. Studies among patients with MD based on health administrative databases found associations between high ED use and diagnosed personality disorders [14], schizophrenia [2], bipolar disorders [2] and a history of ED use [2,14,19] as well as previous hospitalizations [2,19]. Moreover, compared to individuals with lower/no ED use, studies on patients with high one-year ED use identified more anxiety disorders [20,21], depressive disorders [22] and psychotic disorders other than schizophrenia [23], also finding that they made greater use of outpatient health services and residential crisis services [24].

Only three studies to our knowledge, all of them somewhat dated [2,14,19], and from Denmark [2], Switzerland [14] and Finland [19] respectively, investigated recurrent high ED use among patients with MD. Yet none of these studies differentiated among predictors of high ED use over two or three consecutive years, nor were many variables included other than MD diagnoses. A better understanding of which variables predict recurrent high ED use, as opposed to high one-year ED use, may suggest interventions targeted to specific patient profiles that respond more appropriately to their needs. It is also possible that patients with high one-year ED use may display significantly different patterns of care compared with those making high use of ED over two or three consecutive years. This five-year longitudinal study compared predictors of recurrent high ED use over two and three consecutive years with high one-year ED use for a cohort of 3121 high ED users. Based on existing literature, the study hypothesized that recurrent high ED use would be predicted mainly by clinical variables, followed by service use variables and sociodemographic characteristics.

## 2. Materials and Methods 

### 2.1. Study Setting and Data Collection

Data for this longitudinal study were obtained from the Quebec (Canada) Health Insurance Regime (RAMQ) database, which contains medical administrative data including billing files for medical services provided by physicians on a fee-for-service basis, as well as demographic and socioeconomic information on patients. Hospitalization data were obtained from the MED-ECHO database. The Quebec ED database (BDCU) and the public primary health care database (I-CLSC) were other sources used, the BDCU providing information from ED triage nurses on reasons for ED visits and illness acuity, while the I-CLSC contained information on biopsychosocial services dispensed by local community health service centers. The study was approved by the Quebec Access to Information Commission and the ethics committee of a mental health university institute.

For inclusion in the study, patients had to be high ED users, defined as having visited one of six selected ED in Quebec at least three times in 2014–2015 for any medical reason and without high ED use from 2012–2013 to 2013–2014. Patients also had to be at least 12 years old and be diagnosed with MD or SRD from 2012–2013 to their third ED visit in 2014–2015 (ED index date). High ED users were followed from 2014–2015 to 2016–2017 and grouped according to high ED use for one-year (2014–2015), and recurrent high ED use over two or three consecutive years. Patient administrative data were gathered for the two-year period prior to third ED visit in 2014–2015 for purposes of investigating predictors of high ED use, whether recurrent or not. The six ED selected for the study were mainly university hospitals located in urban areas throughout Quebec. 

### 2.2. Variables

The dependent variable, high ED use, included three categories: recurrent high ED use over two years (2014–2015 to 2015–2016), recurrent high ED use over three years (2014–2015 to 2016–2017) and high one-year ED use. A minimum of three visits per year is the agreed definition of high ED use based on previous research [10,22,25], with three or four visits as the usual standard [11]. All study variables are shown in Figure 1, the conceptual framework, which also identifies the databases linked to each variable.

Independent variables were grouped in terms of clinical, sociodemographic, and service use variables. Clinical variables included the following diagnoses: common MD (depressive, anxiety and adjustment disorders); serious MD (schizophrenia spectrum and other psychotic disorders, bipolar disorders) and personality disorders; SRD (alcohol/drug-induced, use, intoxication); and chronic physical illnesses, including severity levels based on the Elixhauser Comorbidity Index [26]. Diagnoses were based on the International Classification of Diseases Ninth Revision (ICD-9) for the RAMQ database and on the Tenth Revision (ICD-10) for the MED-ECHO and BDCU databases (Table 1). Based on the BDCU (ED database), suicidal behaviors (ideation, attempts), violence or social issues as reasons for ED visit, being on a stretcher at the third 2014–2015 ED visit as a proxy for level of patient functionality, and illness acuity at ED visit were also evaluated. Measured with the Canadian Triage Acuity Scale [27] to determine ED treatment priority, Illness acuity ranges from levels 1–2 (immediate or very urgent care), to 3 (urgent care) and 4–5 (less urgent or non-urgent care). Patients assessed at triage levels 4–5 are considered more appropriate for ambulatory care than treatment at ED [27]. 

Sociodemographic variables included sex, age (grouped), and scores on the Material or Social Deprivation Indexes [28]. Based on Canadian census data (2011), these indices are assigned to each dissemination area, the smallest area for which census data are available (400–700 persons), with each area corresponding to a postal code. Material deprivation was calculated based on the proportion of persons in any given neighborhood without a high school diploma, ratio of employed persons to total population and average personal income, while social deprivation based on proportions of single-parent families, persons living alone and those separated, divorced or widowed. These indices were regrouped into quintiles from least (1–2), average (3) to highest (4–5 or not assigned) levels of deprivation. As individuals with no assigned postal code mainly represented residents of public long-term health care facilities or homeless individuals, this subgroup was included in the group with the highest level of material or social deprivation. 

Service use variables included having a family physician, number of consultations with the GP most frequently consulted, number of consultations with the outpatient psychiatrist most frequently consulted, continuity of medical care, number of interventions in local community health service centers (excluding GP consultations) and prior hospitalizations for any medical reason. The most frequent GP consulted, as a proxy for patient family physician, included a minimum of two consultations with the same GP or with two GP working in the same family medicine group [29]. For psychiatrist most frequently consulted, if patients had only one consultation, they must have had at least two consultations with their GP in ambulatory care, representing collaborative care [30]. Having three or more consultations with the same GP or psychiatrist in a 12-month period, particularly in the first three months which represents the acute phase of a MD, is an important indicator for continuity of care [31,32]. Continuity of care was based on the Usual Provider Continuity Index, which describes the proportion of visits to the most frequently consulted GP and psychiatrist of total visits to GP and psychiatrists in ambulatory care [30]. A benchmark of ≥0.8 was selected as high continuity of care [33]. 

### 2.3. Analysis

Descriptive analyses were performed including two-way frequency tables for each independent variable in relation to high one-year ED use, or recurrent high ED use for two or three consecutive years. Independent variables included dummy variables (e.g., depressive disorders, sex) and ordinal variables (e.g., number of outpatient psychiatric consultations with usual medical provider, Social Deprivation Index). Missing values were less than 1%, and complete case analysis was used. No outliers were identified. The intraclass correlation coefficient (ICC) was small (0.047), indicating that correlation among patients in hospital settings was low, and multilevel analysis was not needed. Collinearity statistics were tested using variance inflation factors (VIF) and tolerance tests, with 5 as the maximum level of VIF. Independent variables identified without collinearity were entered into the multinomial logistic regression model with an alpha value of *p* < 0.20. Several models were tested, including all variables, but they didn’t have any effects on the final results. The reference category for the multinomial regression was high ED use over a one-year period. Two odds ratios were calculated for each independent variable: two or three consecutive years of high ED use versus one-year high ED use (95% confidence intervals). Akaike’s Information Criterion (AIC) and the Bayesian information criterion (BIC) were used as the criteria for model selection. Forward stepwise selection was adopted for the estimation of parameters of the multinomial logistic regression. All analyses were performed using SPSS 24.0 [34].

## 3. Results

Of the 3121 high ED users, 73% (*n* = 2289) were high one-year ED users, 15% (*n* = 468) were high ED users for two consecutive years and 12% (*n* = 364) for three consecutive years (Table 2). Most patients (87%) had common MD; 59% had serious MD, 29% personality disorders, 33% SRD and 50% chronic physical illnesses, with low severity for 75% of them (index 0). At ED, 45% of patients showed suicidal behaviors, 5% violence or social issues; 51% of cases were classified as less urgent or non-urgent (illness acuity levels 4–5), and 57% were on stretchers at their third 2014–2015 ED visit. Most patients (37%) were 25 to 44 years, 54% were male, 48% lived in the most materially deprived or unassigned areas (4–5), while 67% lived in the most socially deprived or unassigned areas. Prior to third ED visit in 2014–2015, 47% of patients reported having a family physician; 56% had made 3+ consultations with their GP and 50% with their psychiatrist; 19% received high continuity of medical care; 43% had received interventions in local community health service centers and 67% prior hospitalizations for any medical reason. Table 2 presents the bivariate analyses.

Table 3 presents results from the logistic multinomial regression model. Compared with high one-year ED users, both recurrent high two-year (β = 0.367, OR = 1.443, *p* = 0.001) and three-year ED users (β = 0.352, OR = 1.422, *p* = 0.006) had more personality disorders. Patients with anxiety disorders (β = 0.326, OR = 1.385, *p* = 0.009), alcohol-related (β = 0.542, OR = 1.720, *p* = 0.000) or drug-related disorders (β = 0.596, OR = 1.814, *p* = 0.000), chronic physical illnesses (β = 0.549, OR = 1.731, *p* = 0.000), suicidal behaviors (β = 0.604, OR = 1.826, *p* = 0.000) and violence or social issues (β = 0.837, OR = 2.310, *p* = 0.000) were more likely to be recurrent high three-year ED users, than high one-year ED users. Recurrent high three-year ED users lived in more socially deprived areas (4–5) or areas not assigned (β = 0.468, OR = 1.597, *p* = 0.005), compared with high one-year ED users. Over the previous 12 months, more of these patients had consulted their psychiatrists 3+ times (β = 0.377, OR = 1.458, *p* = 0.006), in local community health service centers (β = 0.298, OR = 1.347; *p* = 0.028) and had more hospitalizations (β = 0.309, OR = 1.362, *p* = 0.037) than high one-year ED users. By contrast, these patients were less likely to have high continuity medical care (β = −0.550, OR = 0.577, *p* = 0.002) than high one-year ED users. The model had an acceptable goodness of fit based on Pearson and deviance chi-square statistics (*p* > 0.05).

## 4. Discussion

The proportion of recurrent high ED users in our cohort (27%) was similar to the percentage (22%) reported in a Swiss study that assessed a cohort of high ED users with MD [14]. These results were also in line with those in ED studies involving different populations (Medicaid users [10], high ED users in general [35], ED users with chronic conditions [18]) where 13–34% of high ED users were recurrent high users [10,15,18]. 

Results confirmed the hypothesis that recurrent high ED use would be predicted mainly by clinical variables, followed by service use and sociodemographic variables. However, clinical variables were more likely to predict three consecutive years of recurrent high ED use and, as such, may be considered a key benchmark for improving response to the needs of the most vulnerable patients and reducing ED overcrowding. A US study using a general hospital database of 47,349 patients having visited ED showed similarly that most high ED users over two consecutive years remained so the following year to become a group of chronically high ED users [12]. 

Personality disorders were the only variable in this study that distinguished recurrent high two-year ED users from high one-year ED users. The contribution of personality disorders in predicting recurrent high ED use was previously reported [14]. Patients with personality disorders, notably borderline personality disorders, often have difficulties establishing therapeutic alliances with clinicians [14], which may explain their pattern of recurrent high ED use over two or three consecutive years.

Compared with high one-year ED users, patients with three consecutive years of high ED use had relatively more acute and co-occurring problems. Anxiety disorders characterized by agitation and difficulty dealing with current life situations, exacerbated at times by painful physical symptoms [35], easily explained recurrent high ED use among these patients. Given that access to medical care is not optimal in most countries, as in Quebec [36,37], ED are often used to respond to acute problems like suicidal behaviors, frequent in MD-SRD, and as suggested by one study, might play a stabilizing role or provide a safety net for these patients [38]. That patients visiting ED for violence or social issues were overrepresented among recurrent high three-year ED users seems to confirm the frequency of crisis episodes among these patients and, at least for some, homelessness. The association of MD or SRD with such issues was previously reported as a key driver of ED use [39,40,41].

Co-occurring MD-SRD and chronic physical illnesses were also previously identified as predictors of recurrent high ED use [10,18]. Patients with MD-SRD reportedly have difficulty accessing ambulatory care [42] due to stigmatization issues in particular [43] or adhere poorly to treatment [44], which also explains recurrent high ED use. GP have relatively little interest in treating these patients [45], while few programs exist in Quebec or elsewhere [14] offering integrated treatment for MD-SRD [46]. The prevalence of co-occurring chronic physical illnesses (e.g., liver disease, cardiovascular disease, diabetes) is also high among patients with MD or SRD [14,29,30], another explanation for recurrent high ED use.

Living in areas with the highest social deprivation or areas not assigned, the only sociodemographic variable that predicted recurrent high ED use over three years, reinforced this population as highly vulnerable and lacking social networks. Previous studies on recurrent high ED users in general or those with chronic conditions found that recurrent high ED users were more likely to live alone [15], whereas living with relatives as well as social support in general are known to protect against bad out-comes like high ED use [47] and encourage more outpatient service use [48,49]. Support from relatives may also complement use of public health services in meeting the needs of patients with MD [50], rather than them relying on the ED.

Contrary to expectation, recurrent high three-year ED users were also more likely to have had 3+ consultations with their psychiatrists and have used more psychosocial services in local community health service centers than others. Perhaps this intensity of service use was, however, insufficient or certain interventions inadequate to meet the needs of patients with such complex health and social problems [2], further explaining the persistence of high ED use over multiple years. Inappropriate care may also result from inefficient organization of healthcare services [51], insufficient deployment of best practices [52], treatment refusal by patients in denial or those who voice concerns about stigmatization [53,54]. Moreover, the frequent use of such services may translate into difficulties for the healthcare system in meeting complex needs among vulnerable patients living in the community. 

Studies have also found that expertise in MH among GP was often inadequate for treating co-occurring MD-SRD [55] or MD-SRD-chronic physical illnesses [56,57]. Moreover, with brief consultation the rule in primary care [58], GP often had insufficient time to treat patients with SRD, MD-SRD or personality disorders. Collaborative care is needed for the management of complex health conditions [59,60]. As well, prior hospitalizations as a predictor for high ED use over three years by patients with MD, reported in our study, has also been reported elsewhere [2,19], confirming the severity of health conditions among these patients [19]. An original finding of this study was the protective role played by high continuity of medical care in preventing recurrent high three-year ED use. Strong continuity of medical care is a well-known target in all healthcare reforms [61,62], especially for patients with complex health conditions living in disadvantaged social contexts [52,63]. By contrast, good continuity of care is usually associated with better health outcomes [30], contributing to a reduction in high ED use [64] and hospitalizations [65].

Serious MD did not predict recurrent high ED use, which was surprising. This may be a result of previous Quebec mental health reforms that mandated the reinforcement of programs such as assertive community treatment [66,67] or intensive case management [66,67] targeting patients with serious MD or with high rates of ED use and hospitalizations or may reflect the more recent implementation of innovative programs involving home treatment teams for serious MD [68]. These programs are known to be effective in reducing ED use and hospitalizations [66,67]. 

This study has certain limitations. First, health administrative databases were primarily developed for financial purposes, not for research and, as such, our data may only be used as proxy measures for patient needs. Second, the data excluded consultations with healthcare providers other than physicians in hospital settings as well as psychosocial services offered in private clinics (e.g., psychologists) and in the voluntary sector (e.g., crisis centers). Yet, these services may have affected rates of recurrent high ED use. At the same time, we would not expect that high ED users would consult very extensively in private settings like the offices of psychologists, whose services are not reimbursed by the government. Moreover, this study did not consider more than 3 consecutive years of recurrent high ED use. A longer follow-up period may have provided other key benchmarks using other distinct ED predictors. Finally, results of this study may not be generalizable to other healthcare systems, notably those without universal coverage, to ED in semi-urban or rural areas or to specialized psychiatric ED.

## 5. Conclusions

This study was one of few to compare predictors of recurrent high ED use over two or three consecutive years with high one-year ED use. The integration of service use and some clinical variables not previously assessed was another important contribution of the study. Results confirmed that recurrent high three-year ED users had more co-occurring and complex health and social problems as compared with high one-year ED users. However, the latter group did not differ substantially from recurrent high two-year ED users. Only having more personality disorders distinguished high two-year ED users from high one-year ED users. Thus, three consecutive years of high ED use may be the appropriate benchmark for targeting high recurrent ED users, whose needs may, in turn, suggest a starting point for improving ambulatory care and reducing high ED use. Apart from personality disorders, recurrent high three-year ED users also experienced more anxiety, alcohol or drug-related disorders, chronic physical illnesses, suicidal behaviors and violence or social issues. They were more frequent users of outpatient psychiatrists, local community health service centers and more had been previously hospitalized, attesting to their complex health profile and overall vulnerability. Yet these patients did not benefit from high continuity of medical care. Considering this, high continuity and more adequate care is needed to meet the complex and multiple needs of these high ED users, including best practices like assertive community treatment, integrated MD-SRD treatment, or home-based treatment. Screening, brief intervention, including motivational treatment, and referral to services, especially SRD services or specialized anxiety or personality disorder clinics, should also be reinforced. As well, services that favor social network development such as self-help groups or day centers should be prioritized for persons affected by loneliness and living in socially deprived neighborhoods. Finally, further research with a more extended period than three consecutive years of high recurrent ED use may be planned to study how results may vary with significantly different predictors. 

## Figures and Tables

**Figure 1 ijerph-18-04559-f001:**
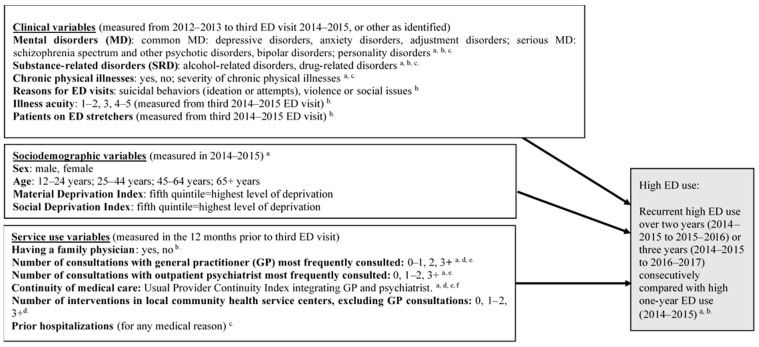
Conceptual framework: predictors of recurrent high emergency department (ED) use. ^a^
*Régie de l’assurance maladie du Québec* (RAMQ—Quebec health and social services database); ^b^
*Banque de données communes des urgences* (BDCU—database for ED visits); ^c^
*Maintenance et exploitation des données pour l’étude de la clientèle hospitalière* (MED-ÉCHO—hospitalization database); ^d^
*Système d’information sur la clientele et les services des CSSS*—*mission CLSC* (I-CLSC—local community health service centers database); ^e^ Most frequently consulted GP (proxy for “patient family physician”) included a minimum of two consultations with the same GP or with two GP working in the same family medicine group. For psychiatrist most frequently consulted, if a patient had only one psychiatrist consultation, he/she must have had at least two consultations with his/her GP in ambulatory care. ^f^ Usual Provider Continuity Index describes the proportion of visits to the most frequently consulted GP and psychiatrist of total visits to GP and psychiatrists in ambulatory care. This index is ranked low (<0.80) or high (≥0.80).

**Table 1 ijerph-18-04559-t001:** Codes for mental disorders (MD) and substance-related disorders (SRD) according to the International Classification of Diseases, Ninth and Tenth revisions ^a^.

Diagnoses	International Classification of Diseases, Ninth Revision (ICD-9)	International Classification of Diseases, Tenth Revision, Canada (ICD-10-CA)
Schizophrenia spectrum and other psychotic disorders	295, 297, 298	F20, F21, F22, F23, F24, F25, F28, F29, F32.3, F33.3, F44.89
Bipolar disorders	296.0–296.6, 296.8, 296.9	F30.0–F30.2, F30.8, F30.9, F31.0–F31.9
Depressive disorders	300.4, 311.9	F32.0- F32.3, F32.8, F32.9, F33.0–F33.3, F33.8, F33.9, F34.8, F34.9, F38.0, F38.1, F38.8, F39, F41.2
Anxiety disorders	300 (except 300.4)	F40–F48, F68
Personality disorders	301	F60, F07.0, F34.0, F341, F48.8, F61
Adjustment disorders	309.0–309.4, 309.8, 309.9	F43.0–F43.2, F43.8, F43.9
Alcohol-related disorders	303.0, 303.9, 305.0 (alcohol abuse or dependence); 291, 357.5, 425.5, 535.3, 571.0–571.3 (alcohol-induced disorders); 980.0, 980.1, 980,8, 980.9 (alcohol intoxication)	F10.1, F10.2 (alcohol abuse or dependence); F10.3–F10.9, K70.0–K70.4, K70.9, G62.1, I42.6, K29.2, K85.2, K86.0, E24.4, G31.2, G72.1, O35.4 (alcohol-induced disorders); F10.0, T51.0, T51.1, T51.8, T51.9 (alcohol intoxication)
Drug-related disorders	304.0–304.9, 305.2–305.7, 305.9 (drug abuse or dependence); 292 (drug-induced disorders); 965.0, 965.8, 967.0, 967.6, 967.8, 967.9, 969.4–969.9, 970.8, 982.0, 982.8 (drug intoxication)	F11.1, F12.1, F13.1, F14.1, F15.1, F16.1, F18.1, F19.1, F11.2, F12.2, F13.2, F14.2, F15.2, F16.2, F18.2, F19.2, F55 (drug abuse or dependence); F11.3–F11.9, F12.3–F12.9, F13.3–F13.9, F14.3–F14.9, F15.3–F15.9, F16.3–F16.9, F18.3–F18.9, F19.3–F19.9 (drug-induced MD); F11.0, F12.0, F13.0, F14.0, F15.0, F16.0, F18.0, F19.0, T40.0–T40.9, T42.3, T42.4, T42.6, T42.7, T43.5, T43.6, T43.8, T43.9, T50.9, T52.8, T52.9 (drug intoxication)

^a^ MD and SRD identified in RAMQ (Régie de l’assurance-maladie du Québec) were based on the International Classification of Diseases Ninth Revision (ICD-9), while MED-ÉCHO (Hospitalization database: Maintenance et exploitation de données pour l’étude de la clientèle hospitalière) and BDCU (Emergency department database: Banque de données communes des urgences) were based on the ICD Tenth Canadian Revision (ICD-10-CA).

**Table 2 ijerph-18-04559-t002:** Characteristics of one-year high emergency department (ED) users (3+ ED visits/2014–2015 (index year)) versus recurrent high ED users over two or three consecutive years.

Characteristics Overall	Total High ED Users in 2014–2015 (Index Year)	High One-Year ED Users	Recurrent High ED Users for Two Consecutive Years (2014–2015 and 2015–2016)	Recurrent High ED Users for Three Consecutive Years (2014–2015 to 2016–2017)	*p*-Value
	*N* (%)	*N* (%)	*N* (%)	*N* (%)	
	3121 (100)	2289 (100)	468 (100)	364 (100)	
**Clinical variables (2012–2013 to third ED visit in 2014–2015) or other, as specified**					
Mental disorders (MD) ^a^					
Common MD					
Depressive disorders	1510 (48.4)	1114 (48.7)	220 (47.0)	176 (48.4)	0.807
Anxiety disorders	1740 (55.8)	1239 (54.1) ^3^	260 (55.6) ^3^	241 (66.2) ^1,2^	**0.000**
Adjustment disorders	1347 (43.2)	983 (42.9)	207 (44.2)	157 (43.1)	0.877
Serious MD					
Schizophrenia spectrum and other psychotic disorders	1292 (41.4)	926 (40.5)	209 (44.7)	157 (43.1)	**0.188**
Bipolar disorders	1169 (37.5)	850 (37.1)	168 (35.9)	151 (41.5)	0.212
Personality disorders	913 (29.3)	600 (26.2) ^2,3^	161 (34.4) ^1,3^	152 (41.8) ^1,2^	**0.000**
Substance-related disorders (SRD)					
Alcohol-related disorders	715 (22.9)	478 (20.9) ^3^	116 (24.8) ^3^	121 (33.2) ^1,2^	**0.000**
Drug-related disorders	573 (18.4)	373 (16.3) ^2,3^	94 (20.1) ^1,3^	106 (29.1) ^1,2^	**0.000**
Chronic physical illnesses	1563 (50.1)	1089 (47.6) ^3^	244 (52.1) ^3^	230 (63.2) ^1,2^	**0.000**
Elixhauser Comorbidity Index ^b^					**0.025**
0	2331 (74.7)	1738 (75.9)^3^	350 (74.8)	243 (66.8) ^1^	
1	315 (10.1)	222 (9.7)	44 (9.4)	49 (13.5)	
2	200 (6.4)	138 (6.0)	32 (6.8)	30 (8.2)	
3+	275 (8.8)	191 (8.3)	42 (9.0)	42 (11.5)	
Reasons for emergency department (ED) visits					
Suicidal behaviors (ideation or attempts)	1401 (45.0)	973 (42.6) ^3^	216 (46.2) ^3^	212 (58.4) ^1,2^	**0.000**
Violence or social issues	168 (5.4)	109 (4.8) ^3^	21 (4.5) ^3^	38 (10.4) ^1,2^	**0.000**
Illness acuity (triage priority levels) ^c^					0.154
Level 1–2 (immediate or very urgent care)	525 (16.8)	403 (17.6)	66 (14.1)	56 (15.4)	
Level 3 (urgent care)	1000 (32.0)	742 (32.4)	141 (30.1)	117 (32.1)	
Level 4–5 (less urgent or non-urgent care)	1596 (51.1)	1144 (50.0)	261 (55.8)	191 (52.5)	
ED patients on stretchers ^c^	1780 (57.0)	1332 (58.2) ^3^	258 (55.1)	190 (52.2) ^1^	0.067
**Sociodemographic variables (2014–2015)**					
Age					0.750
12–24 years	665 (21.3)	478 (20.9)	98 (20.9)	89 (24.5)	
25–44 years	1166 (37.4)	852 (37.2)	182 (38.9)	132 (36.3)	
45–64 years	887 (28.4)	661 (28.9)	131 (28.0)	95 (26.1)	
65+ years	403 (12.9)	298 (13.0)	57 (12.2)	48 (13.2)	
Sex					0.174
Male	1683 (53.9)	1221 (53.3)	249 (53.2)	151 (41.5)	
Female	1438 (46.1)	1068 (46.7)	219 (46.8)	213 (58.5)	
Material Deprivation Index					0.666
1–2: least deprived	1060 (34.0)	788 (34.4)	151 (32.3)	121 (33.2)	
3	570 (18.3)	426 (18.6)	82 (17.5)	62 (17.0)	
4–5: most deprived or not assigned ^d^	1491 (47.8)	1075 (47.0)	235 (50.2)	181 (49.8)	
Social Deprivation Index					**0.039**
1–2: least deprived	615 (19.7)	469 (20.5) ^3^	92 (19.7)	54 (14.8) ^1^	
3	409 (13.1)	312 (13.6)	56 (12.0)	92 (19.7)	
4–5: most deprived or not assigned ^d^	2097 (67.2)	1508 (65.9)^3^	320 (68.4)	269 (73.9) ^1^	
**Service use variables (12 months prior to third ED visit in 2014–2015)**					
Having a family physician	1469 (47.1)	1067 (46.7)	223 (47.6)	179 (49.3)	0.632
Number of consultations with general practitioner (GP) most frequently consulted ^e^					0.894
0 consultations	634 (20.3)	464 (20.3)	97 (20.7)	73 (20.1)	
1–2 consultations	731 (23.4)	538 (23.5)	114 (24.4)	79 (21.7)	
3+ consultations	1756 (56.3)	1287 (56.2)	257 (54.9)	212 (58.2)	
Number of consultations with psychiatrist most frequently consulted ^f^					**0.002**
0 consultations	1193 (38.2)	913 (39.9) ^3^	174 (37.2)	120 (33.0) ^1^	
1–2 consultations	374 (12.0)	274 (12.0)	57 (12.2)	43 (11.8)	
3+ consultations	1554 (49.8)	1102 (48.1) ^3^	237 (50.6)	201 (55.2) ^1^	
Usual Provider Continuity Index integrating GP and psychiatrist ^g^					
≥0.80	580 (18.6)	451 (19.7) ^3^	87 (18.6) ^3^	42 (11.5) ^1,2^	**0.001**
Number of interventions in local community health service centers (excluding GP interventions)					**0.001**
0 interventions	1784 (57.2)	1337 (58.4) ^3^	272 (58.1) ^3^	175 (48.1) ^1,2^	
1–2 interventions	600 (19.2)	444 (19.4)	79 (16.9)	77 (21.2)	
3+ interventions	737 (23.6)	508 (22.2) ^3^	117 (25.0)	112 (30.8) ^1^	
Prior hospitalizations (any medical reason)	2082 (66.7)	1479 (64.6) ^3^	321 (68.6) ^3^	282 (77.5) ^1,2^	**0.000**

^a^ Patients may have more than one mental disorder—total percentage may exceed 100%. ^b^ Chronic physical illnesses included: chronic pulmonary disease, cardiac arrhythmias, tumor w/o metastasis, renal disease, fluid electrolyte disorders, myocardial infarction, congestive heart failure, metastatic cancer, dementia, stroke, neurological disorders, liver disease (excluding alcohol-induced liver disease), pulmonary circulation disorders, coagulopathy, weight loss, paralysis, AIDS/HIV. ^c^ Measured at the third 2014–2015 ED visit. ^d^ Missing address or living in an area where index assignment is not available. An index cannot usually be assigned to residents of long-term health care units or homeless individuals. ^e^ The most frequent GP consulted (proxy of the “patient family physician”) needs to include a minimum of two consultations with the same GP or with at least two different GP working in a same family medicine group. ^f^ For the psychiatrist most frequently consulted, if a patient had only one psychiatrist consultation, he/she must have had at least two consultations with his or her GP in ambulatory care. ^g^ Usual Provider Continuity Index describes the proportion of visits to the most frequently consulted GP and psychiatrist of total visits to GP and psychiatrists in ambulatory care. This index is ranked low (<0.80) or high (≥0.80). χ^2^ Comparisons were provided for each row reporting percentages for categorical variables. Superscript and bold indicated significant differences at *p* < 0.05 between the three categories of the dependant variable (high ED use over a one-year (1)**,** recurrent high ED use for two consecutive years (2) and recurrent high ED use for three consecutive years (3)).

**Table 3 ijerph-18-04559-t003:** Multinomial logistic regression analysis on predictors of recurrent high emergency department (ED) use over two or three consecutive years as compared with high ED use over a one-year period.

		Recurrent High ED Use for Two Consecutive Years (2014–2015 and 2015–2016)					Recurrent High ED Use for Three Consecutive Years (2014–2015 to 2016–2017)			
Variables	β	*p*-Value	OR	95% CI		β	*p*-Value	OR	95% CI	
Clinical variables (2012–2013 to third ED visit in 2014–2015, or other as specified) ^a,b^										
Mental disorders (MD)										
Common MD										
Anxiety disorders	0.020	0.845	1.021	0.833	1.251	0.326	**0.009**	1.385	1.085	1.768
Serious MD										
Personality disorders	0.367	**0.001**	1.443	1.153	1.805	0.352	**0.006**	1.422	1.107	1.827
Substance-related disorders (SRD)										
Alcohol-related disorders	0.155	0.221	1.167	0.911	1.495	0.542	**0.000**	1.720	1.326	2.230
Drug-related disorders	0.170	0.214	1.186	0.906	1.551	0.596	**0.000**	1.814	1.379	2.387
Chronic physical illnesses	0.168	0.124	1.182	0.955	1.463	0.549	**0.000**	1.731	1.347	2.223
Reasons for emergency department (ED) visits										
Suicidal behaviors (ideation or attempts)	0.120	0.263	1.128	0.914	1.392	0.604	**0.000**	1.826	1.433	2.335
Violence or social issues	−0.047	0.852	0.955	0.586	1.555	0.837	**0.000**	2.310	1.505	3.544
ED patients on stretchers ^c^	−0.181	0.217	0.835	0.627	1.112	−0.177	0.357	0.838	0.575	1.221
Sociodemographic variables (2014–2015)										
Social Deprivation Index (ref. 1–2: least deprived)										
3	−0.076	0.682	0.927	0.644	1.334	0.154	0.498	1.167	0.747	1.823
4–5: most deprived or not assigned ^d^	0.098	0.458	1.102	0.852	1.426	0.468	**0.005**	1.597	1.155	2.208
Service use variables (12 months prior to third ED visit)										
Number of outpatient psychiatric consultations with usual medical provider; (ref.: 0 consultations) ^e^										
1–2 consultations	0.037	0.834	1.036	0.742	1.448	0.134	0.514	1.143	0.765	1.707
3+ consultations	0.036	0.748	1.038	0.827	1.303	0.377	**0.006**	1.458	1.114	1.908
Continuity of medical care: Usual Provider Continuity Index integrating GP and psychiatrist (≥0.8) ^f^	−0.045	0.738	0.956	0.735	1.243	−0.550	**0.002**	0.577	0.405	0.821
Number of interventions in local community health service centers, excluding GP consultations; (ref.: 0)										
1–2 interventions	−0.242	0.109	0.785	0.584	1.055	0.027	0.872	1.027	0.743	1.419
3+ interventions	0.048	0.639	1.049	0.828	1.328	0.298	**0.028**	1.347	1.033	1.757
Prior hospitalizations (for any medical reason)	0.134	0.267	1.144	0.902	1.450	0.309	**0.037**	1.362	1.018	1.823

^a^ Patients may have more than one mental disorder—total percentage may exceed 100%. ^b^ The reference group is those having answered “no” for each clinical variable. ^c^ Measured from third 2014–2015 ED visit. ^d^ Missing address or living in an area where index assignment is not available. An index cannot usually be assigned to residents of long-term health care units or homeless individuals. ^e^ For most frequently consulted psychiatrist (ambulatory care only), if a patient had only one consultation, he/she must have at least two consultations with his/ her GP. ^f^ Usual Provider Continuity Index describes the proportion of visits to the most frequent consulted GP and psychiatrist of total visits to GP and psychiatrists in ambulatory care. The most frequently consulted GP (proxy for “patient family physician”) includes a minimum of two consultations with the same GP or with two GP working in the same family medicine group. Bold indicated significant differences at *p* < 0.05 for independent variables in multinomial regression.

## Data Availability

In accordance with the applicable ethics regulations in the province of Quebec, the authors do not have permission to share the data extracted for this study from the Quebec Health Insurance Board (RAMQ) database.

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
