# Peer review of "Predictors of Recurrent High Emergency Department Use among Patients with Mental Disorders"

_ijerph, 2021, doi:10.3390/ijerph18094559_

Round 1

Reviewer 1 Report

The article “ Predictors of recurrent high emergency department use among patients with mental disorders” presents valuable data on the topic, going beyond previous research by exploring the specific predictions that differ in higher degrees of recurrence in emergency department among patients with mental disorders.

The authors state this clearly and refer to relevant and updated research on the topic.

I recommend the publication of this article after revising a few important issues (see below).

#Introduction
Lines 33-59

The authors present updated research on the topic, yet the references to the type of research and populations involved are quite vague and may be misleading. Please consider revising this section with more detail, indicating at least the context and type of research supporting the empirical evidence.

For example: in lines 35-38 there are made generic considerations on ED users, yet the reference is from a systematic review study that includes only USA populations; in lines 38-39, you mention “In the US….” yet the study refers to NY population (not national representative).

Of course, generic consideration can be made based on the consulted studies, but when choosing to present percentages or addressing specific associations it is important to inform the reader from where they came from and how representative they are for the population of adults with MD.

#Hipothesys
Lines 69-71

Please revise “more strongly predicted.” This is a strange formulation because implies that you are testing the strength or magnitude of effects, but you don’t refer to any analytical strategy for this (for example comparing power or accuracy in different models w different variables types). Maybe rephrased has “Based on existing literature, the study hypothesized that recurrent high ED use will be predicted mostly by clinical variables, followed by service use variables and sociodemographic characteristics.”

Or, instead, add/clarify your empirical rationale (and additional results and discussion) on how you test the differences predicting power between variables types.

#2.2. Variables
Lines 92-126

Please revise the section to make clear:

-             If all variables are included in the regression models as categorical variables (this because, a few could be considered ordinal in the models, also the authors could precoded the variables as dummies before data analysis, for example). Please clarify after presenting variables how they are consider in the models.
-             Include “ED patients on stretchers” variable  in the description (it is not mentioned here but then it appears on data tables)
-             Lines 128- 134 Not clear if you collect data on Material or Social Deprivation Indexes from somewhere or if you calculated yourself with census data, please clarify. Also, include here the justification to assign the highest level of deprivation to people with not code post assigned (as you do in footnotes)
-             Lines 141 Add info on how you grouped the Number of consultations of GP and Psy (3 categories...)

# Analysis
Lines 145-155

Please revise the section to make clear:

  • How do you treated outliers (logistic regressions are very sensible to outliers)

  • How do you select variables for your regression model (it seems that you use bivariate analysis to select the most relevant variables to the model, but this is not stated anywhere)

#Results
Correct headings of table 2! 

Line 169
Add bivariate analysis description. Which variables are associated to ED high use level? Re-state how this informs your regression model (variable selection).

Lines 184-193
Please revise the section:
- The presentation of the results is too synthetic and does not report to the statistical indicators. Please prevision, and add in parenthesis betas, OR and p-value of each relevant effect (b=, OR=, p=). OR are particularly illustrative of the relative changes between the categories in comparison (example: average chances of having drug-related disorders almost doubles among higher ED use) and can be also referred as a relative percentage increase/decrease.
- Deviance and Qui-Square statistics on the global quality of the regression model are missing.

Lines 206-207
Please revise and/or clarify:
“clinical variables would account most strongly for recurrent high ED use”
This is a strange formulation because you are not comparing the magnitude of effects, only which type of predicter have or not a relevant effect. You should rephrase for sake of clarity, saying most of the relevant effects are clinical variables as expected (see comments above on hypothesis).

#Discussion
Line 213-272

Please revise referring to studies populations when relevant, for exampling the country from where the population is (as commented before in Introduction section).

Other minor issues should be addressed across the text.
Please revise the manuscript in order to be consistent in details such:  abbreviations use and abbreviation descriptions (sometimes descriptions are capitalised others not), french excerpts/names in italic (sometimes are in italic and others not).
Additionally, please improve tables readability: omit vertical lines and unnecessary horizontal lines (in tables 2 and 3, between categories of the same variable for example); consider putting in bold coefficients with relevant associations (p<.05). 

Author Response

Open Review

Comments and Suggestions for Authors

The article “ Predictors of recurrent high emergency department use among patients with mental disorders” presents valuable data on the topic, going beyond previous research by exploring the specific predictions that differ in higher degrees of recurrence in emergency department among patients with mental disorders.

The authors state this clearly and refer to relevant and updated research on the topic.

I recommend the publication of this article after revising a few important issues (see below).

Response: Thank you very much!

#Introduction
Lines 33-59

The authors present updated research on the topic, yet the references to the type of research and populations involved are quite vague and may be misleading. Please consider revising this section with more detail, indicating at least the context and type of research supporting the empirical evidence.

For example: in lines 35-38 there are made generic considerations on ED users, yet the reference is from a systematic review study that includes only USA populations; in lines 38-39, you mention “In the US….” yet the study refers to NY population (not national representative).

Of course, generic consideration can be made based on the consulted studies, but when choosing to present percentages or addressing specific associations it is important to inform the reader from where they came from and how representative they are for the population of adults with MD.

Response: We have provided more information about countries where studies were conducted, types of samples (ex: Medicaid users, general population, or patients with mental disorders) and sources used (administrative data, survey).

#Hipothesys
Lines 69-71

Please revise “more strongly predicted.” This is a strange formulation because implies that you are testing the strength or magnitude of effects, but you don’t refer to any analytical strategy for this (for example comparing power or accuracy in different models w different variables types). Maybe rephrased has “Based on existing literature, the study hypothesized that recurrent high ED use will be predicted mostly by clinical variables, followed by service use variables and sociodemographic characteristics.”

Response: Correction was done as suggested! Thank you!

Or, instead, add/clarify your empirical rationale (and additional results and discussion) on how you test the differences predicting power between variables types.

#2.2. Variables
Lines 92-126

Please revise the section to make clear:

Response: This section was revised.

-             If all variables are included in the regression models as categorical variables (this because, a few could be considered ordinal in the models, also the authors could precoded the variables as dummies before data analysis, for example). Please clarify after presenting variables how they are consider in the models.

Response: Thank you for this comment. We now indicate in the analysis section how variables were included in the model. Independent variables including dummy variables (e.g.: depressive disorders, sex) and ordinal variable (e.g.: number of out-patient psychiatric consultations with usual medical provider, Social Deprivation Index…).

-             Include “ED patients on stretchers” variable in the description (it is not mentioned here but then it appears on data tables)

Response: As indicated in the description of independent variables (methods), being on a stretcher at ED is a proxy for level of patient functionality. We tested this variable in the multivariate model, but it was not significant, which is why we don’t talk about this variable in the results section. We have now included this variable at least in the descriptive analysis of the results section of the article. The variables for the index year (2014-15) were calculated based on the third ED visit.

-             Lines 128- 134 Not clear if you collect data on Material or Social Deprivation Indexes from somewhere or if you calculated yourself with census data, please clarify. Also, include here the justification to assign the highest level of deprivation to people with not code post assigned (as you do in footnotes)

Response: Scores for Material and Social deprivation indices were not calculated by our team but are systematically included in the Quebec health and social services databank. These indices were created at the end of the year 1990 with the aim of measuring deprivation among Quebecers and Canadians at the smallest area code levels. As indicated in the footnote under Table 2, and now repeated in the text, individuals with no postal code include residents of public long-term health care facilities or homeless individuals, which justifies their assignment in the group with the highest material or social deprivation.

-             Lines 141 Add info on how you grouped the Number of consultations of GP and Psy (3 categories...)

Response: In the description of services, we included information about GP and psychiatrists more frequently consulted, as well as in a footnote to Table 2. The choice of three consultations with the same GP or psychiatrist was created as a distinct category, because three consultations or more within a 12-month period, and particularly in the first 3 months representing the acute phase of mental disorders, is an important indicator of continuity of care. References were included in this section.

# Analysis
Lines 145-155

Please revise the section to make clear:

Response: The section was revised as suggested.

  • How do you treated outliers (logistic regressions are very sensible to outliers)

Response: Thank for your comment, but we did not have any outliers, all variables were studied in descriptive analyses (distribution, frequency) before producing the logistic regression.

  • How do you select variables for your regression model (it seems that you use bivariate analysis to select the most relevant variables to the model, but this is not stated anywhere)

Response: In the analyses, we tested several possible models: all variables and only with significant results in bivariate analyses (p=0.20). The stepwise method was used for estimation of parameters for the multinomial regression.

#Results
Correct headings of table 2! 

Response: Corrections were done. Thanks!

Line 169
Add bivariate analysis description. Which variables are associated to ED high use level? Re-state how this informs your regression model (variable selection).

Response: We included superscripts in Table 1 indicating variables associated with high levels of ED use and in the analysis section, criteria for model selection and the method adopted for estimation of parameters.

Lines 184-193
Please revise the section:
- The presentation of the results is too synthetic and does not report to the statistical indicators. Please prevision, and add in parenthesis betas, OR and p-value of each relevant effect (b=, OR=, p=). OR are particularly illustrative of the relative changes between the categories in comparison (example: average chances of having drug-related disorders almost doubles among higher ED use) and can be also referred as a relative percentage increase/decrease.
- Deviance and Qui-Square statistics on the global quality of the regression model are missing.

Response: We included the statistical indicators (b=, OR=, p=) and goodness of fit in Table 3 and in the results section.

Lines 206-207
Please revise and/or clarify:
“clinical variables would account most strongly for recurrent high ED use”
This is a strange formulation because you are not comparing the magnitude of effects, only which type of predicter have or not a relevant effect. You should rephrase for sake of clarity, saying most of the relevant effects are clinical variables as expected (see comments above on hypothesis).

Response: We have revised the sentence to be similar to the statement of the hypothesis in the introduction. Thank you for this!

#Discussion
Line 213-272

Please revise referring to studies populations when relevant, for exampling the country from where the population is (as commented before in Introduction section).

Response: We have provided more information on countries or types of populations assessed in the referenced studies.

Other minor issues should be addressed across the text.
Please revise the manuscript in order to be consistent in details such:  abbreviations use and abbreviation descriptions (sometimes descriptions are capitalised others not), french excerpts/names in italic (sometimes are in italic and others not).

Response: Verification and corrections were made. Thanks!

Additionally, please improve tables readability: omit vertical lines and unnecessary horizontal lines (in tables 2 and 3, between categories of the same variable for example); consider putting in bold coefficients with relevant associations (p<.05). 

Response: We modified Table 2, omitting vertical and horizontal lines. Coefficients were input in bold in Table 2 and Table 3.

Thank you very much for your comments and suggestions!

Reviewer 2 Report

This paper is well written, and the presentation is good. Below are some suggestions for improvement.

1. Given that the codes in Table 1 do not appear in the presentation of analyses and results of this paper, I think this table does not provide any additional understanding to the reader. Hence, I suggest that Table 1 be removed.

2. Line 143: “…GP and psychiatrists in ambulatory care. [28].” Please remove full stop before “[28]”.

3. Line 150/151: “Independent variables found without collinearity were entered into the multinomial logistic regression model….”

I would presume that removing a few of the correlated variables will change the VIF values for the remainder of the correlated variables, possibly, reducing the VIF values and suggesting that their inclusion may not hurt the prediction model. If so, shouldn't there be a more efficient way of selecting the predictors rather simply throwing out all collinear variables?

Given that the importance of these deleted variables is not discussed, it makes one wonder whether the deletion is indeed of no effect on the result of this paper.

4. Line 158/159: “Of the 3,121 high ED users, 73% (n=2,289) were high one-year ED users, 15% (n=468) were high ED users for two consecutive years and 12% (n= 364) for three consecutive years (Table 2).”

Values from Table 2 (3rd and 5th columns) read “2,289 were recurrent high ED users for three consecutive years (2014-15 to 2016-17) and 364 were high ED users in 2014-15.” This presentation contradicts the statement of results in Line 158/159 as quoted above. Hence, the presentation of results on Table 2 does not align with the statements under Results section.

5. Line 290/291: “Thus, three consecutive years of high ED use may be the 290 appropriate benchmark for targeting high recurrent ED users….”

This study did not consider 4 or more consecutive years. Is it possible that such consideration could shift the benchmark from 3 years that was reported as appropriate?

Moreover, could there be a pattern recurrence observable over longer years (possibly longer than three years)? A possible interest is to investigate how the recurrence pattern changes over time. Such insight may be very important for long-term planning.

Author Response

This paper is well written, and the presentation is good. Below are some suggestions for improvement.

Response: Thank you very much for your comments and suggestions!

  1. Given that the codes in Table 1 do not appear in the presentation of analyses and results of this paper, I think this table does not provide any additional understanding to the reader. Hence, I suggest that Table 1 be removed.

Response: Although Table 1 is not essential, we prefer to keep it rather than adding information under the other tables or directly in the text (which would be quite long) – hoping this is not an issue for you!

  1. Line 143: “…GP and psychiatrists in ambulatory care. [28].” Please remove full stop before “[28]”.

Response: Done!

  1. Line 150/151: “Independent variables found without collinearity were entered into the multinomial logistic regression model….”

I would presume that removing a few of the correlated variables will change the VIF values for the remainder of the correlated variables, possibly, reducing the VIF values and suggesting that their inclusion may not hurt the prediction model. If so, shouldn't there be a more efficient way of selecting the predictors rather simply throwing out all collinear variables?

Given that the importance of these deleted variables is not discussed, it makes one wonder whether the deletion is indeed of no effect on the result of this paper.

Response: Thank you for your comment! We tested several models, including all variables and significant variables only in the bivariate analyses; we didn’t find any differences between models. We also included in the analysis section (methods) of the article the methods adopted for the estimation of parameters.

  1. Line 158/159: “Of the 3,121 high ED users, 73% (n=2,289) were high one-year ED users, 15% (n=468) were high ED users for two consecutive years and 12% (n= 364) for three consecutive years (Table 2).”

Values from Table 2 (3rd and 5th columns) read “2,289 were recurrent high ED users for three consecutive years (2014-15 to 2016-17) and 364 were high ED users in 2014-15.” This presentation contradicts the statement of results in Line 158/159 as quoted above. Hence, the presentation of results on Table 2 does not align with the statements under Results section.

Response: You are correct, there was an error in the titles of columns related to high ED users in 2014-15 (n=2,289) and to recurrent ED users for three consecutive years (n=364). Corrections were done! Thank you for this!

  1. Line 290/291: “Thus, three consecutive years of high ED use may be the 290 appropriate benchmark for targeting high recurrent ED users….”

This study did not consider 4 or more consecutive years. Is it possible that such consideration could shift the benchmark from 3 years that was reported as appropriate?

Response: This is a very pertinent remark, thank you! We have included your comment in the limitations section of the study. However, as shown in the study, there were not many differences between years 1 and 2, contrary to differences for the 3 consecutive years.

Moreover, could there be a pattern recurrence observable over longer years (possibly longer than three years)? A possible interest is to investigate how the recurrence pattern changes over time. Such insight may be very important for long-term planning.

Response: Thank you for this important remark! We have included your comment in the conclusion as another relevant investigation that could be made.

Thank you very much for your comments and suggestions!